

# Assessing the influence of an extended hurricane season on inland flooding potential in the Southeast United States

Monica H. Stone[1], Sagy Cohen[1]

[1]Department of Geography, The University of Alabama, Tuscaloosa, 35487-0322, USA

*Correspondence to*: Monica H. Stone (mhstone@crimson.ua.edu)

**Abstract.** Recent tropical cyclones, like Hurricane Katrina, have been some of the worst the United States has experienced. Tropical cyclones are expected to intensify, bringing about 20 % more precipitation, in the near future in response to global climate warming.  Further, global climate warming may extend the hurricane season.  This study focuses on four major river basins (Neches, Pearl, Mobile, and Roanoke) in the Southeast United States that are frequently impacted by tropical

cyclones. An analysis of the timing of tropical cyclones that impact these river basins found that most occur during the low discharge season, and thus rarely produce riverine flooding conditions.  However, an extension of the current hurricane season of June-November, due to global climate warming, could encroach upon the high discharge seasons in these basins, increasing the susceptibility for riverine hurricane-induced flooding.  This analysis shows that an extension of the hurricane season to May-December (just 2 months longer) increased the number of days that would be at risk to flooding were the

average tropical cyclone to occur by 37-258 %, depending on the timing of the hurricane season in relation to the high discharge seasons on these rivers.  Future research should aim to extend this analysis to all river basins in the United States that are impacted by tropical cyclones in order to provide a bigger picture of which areas are likely to experience the worst increases in flooding risk due to a probable extension of the hurricane season with expected global climate change in the near future.

**1 Introduction**

In the Southeast United States tropical cyclones are some of the most severe rain events (Schumacher and Johnson, 2006).  While tropical cyclones occur less frequently than other rain-producing events, they cause the most damage because they cover a large geographic area and often cause widespread flooding (Greenough et al., 2001; Mousavi, Irish, Frey, Olivera, and Edge, 2011; Schumacher and Johnson, 2006).  On average, tropical cyclones occurring in the Southeast bring



240.4 mm of rain in a 24-hour period (Schumacher and Johnson, 2006). The severity of flooding following tropical cyclone events is a function of tropical storm frequency, landfall location, precipitation intensity, and for coastal areas, mean sea level (Irish and Resio, 2013). In addition to flooding, these storms cause further damage from their strong winds (Greenough et al., 2001; Mousavi et al., 2011), and they frequently can cause tornadoes and landslides (Greenough et al., 2001; NSB,
5    2007).

Coastal communities in the United States, especially along the East Coast and the Gulf Coast, are most at risk to the flooding, strong winds, and heavy precipitation associated with tropical cyclones (Irish, Sleath, Cialone, Knutson, and Jenson, 2014). Unfortunately, approximately half of the United States population lives within only 50 miles of the coast (NSB, 2007), and, on average, areas that are prone to tropical cyclones are 5 times more heavily populated than the rest of
the nation (Frey et al., 2010). Recent increases in coastal populations and development in coastal areas is posing an increasing risk to human life and coastal infrastructure (Greenough et al., 2001; Irish et al., 2014). Though there has been significant growth since, infrastructure along the East and Gulf coasts was worth approximately $3 trillion in the 1990s (NSB, 2007).

Hurricanes are the costliest and cause the most damage of all weather hazards that occur in the United States (Frey et al.,
2010; NSB, 2007). The monetary losses from hurricanes are increasing; in 2006 dollars, average annual losses were $1.3 billion from 1949-1989, $10.1 billion from 1990-1995, and $35.8 billion from 2002-2007 (NSB, 2007). As mentioned above, tropical cyclone events often cause widespread, destructive flooding. Floods lead to more deaths in the United States than any other natural hazard (Greenough et al., 2001; NSB, 2007), and half of the deaths worldwide from natural hazards are due to floods (Schumann and Di Baldassarre, 2010). About 70 million people live in hurricane-prone areas (Greenough
et al., 2001). Flooding from high storm surges during hurricanes has caused approximately 14,600 deaths over the last century; about 50-100 deaths occur per hurricane event (Greenough et al., 2001). In addition to deaths caused by flooding, hurricanes can cause a variety of health impacts including: illnesses that result from ecological changes (changes in the abundance and distribution of disease-carrying insects and rodents, and mold and fungi), damage to healthcare infrastructure and reduced access to healthcare services, damage to water and sewage systems, overcrowded conditions in shelters, and
psychological effects from the trauma faced by victims (Greenough et al., 2001).

Inland areas are also impacted by tropical cyclones; several studies have looked at the influence of tropical cyclones on inland river flooding in small catchments. Kostaschuk, Terry, and Raj (2001) investigated tropical cyclone-induced flooding in the Rewa River system in Viti Levu, Fiji. They observed that rainstorms caused a higher number of floods, but that floods caused by tropical cyclones were much larger (Kostaschuk et al., 2001). Waylen (1991) conducted a partial duration series



flood analysis for the Santa Fe River in Florida, and found similar results. Tropical cyclone-induced floods were found to occur less often than floods from other rain-producing events, however, they tended to have larger magnitudes and longer durations (Waylen, 1991). Specifically, they found that tropical cyclone floods were ~3 times larger and ~2 times longer than other floods (Waylen, 1991). Tropical cyclones bring about 15 % of the precipitation that occurs in the Southeast

during the hurricane season, which is enough to end most droughts that occur in the Southeast (Maxwell, Soulé, Ortegren, and Knapp, 2012).

Numerous studies have indicated that global climate warming may intensify tropical cyclones, and is very likely to result in sea level rise (Bronstert, Niehoff, and Büger, 2002; Frey et al., 2010; Greenough et al., 2001; Irish and Resio, 2013; Irish et al., 2014; Kostaschuk et al., 2001; Mousavi et al., 2011; Ouellet, Saint-Laurent, and Normand, 2012). Majors hurricanes,

those that are Category 3 or higher on the Saffir-Simpson scale, are the most likely to intensify (Frey et al., 2010; Mousavi et al., 2011), however there is some debate about changes in tropical cyclone frequency. Some research predicts that tropical cyclone frequency will increase (e.g. Greenough et al., 2001; Ouellet et al., 2012). This hypothesis was refuted by Irish and Resio (2013) and Kostaschuk et al. (2001); both showed that tropical cyclones are likely to intensify with global climate warming, but occur less frequently.

Greenhouse gases in the atmosphere not only increase atmospheric temperature, but also can lead to increased sea-surface temperatures (Irish et al., 2014). The warmer the sea surface temperature, the more intense tropical cyclones are, thus, global warming may intensify tropical cyclones, such that storms may tend to have higher storm surge levels (Frey et al., 2010; Irish et al., 2014; Mousavi et al., 2011). Between the time periods 1850-1899 and 2001-2005 global sea-surface temperatures rose 0.55 °C (Irish et al., 2014). The Intergovernmental Panel on Climate Change (IPCC) predicts that global

sea-surface temperatures will increase 1.1-6.4 °C over the next century (Irish and Resio, 2013; Mousavi et al., 2011). Sea surface temperatures need to be at or above ~26.7 °C for tropical cyclones to form (Steenhof and Gough, 2008). The current hurricane season extends from June to November, however longer seasons (i.e. storms occurring before June and/or after November) have been occurring in recent years (Dwyer et al., 2015). While research on this topic is not conclusive, there is some indication that global climate change may lead to a change in the Atlantic hurricane season (Dwyer et al., 2015). There

is an 8 % increase in a tropical cyclone's pressure differential for a 1 °C increase in tropical sea-surface temperature (Irish and Resio, 2013; Irish et al., 2014; Mousavi et al., 2011). Further, there is a 3.7 % increase in a tropical cyclone's wind speed for a 1 °C increase in tropical sea-surface temperature (Irish et al., 2014). Climate models also suggest that precipitation rates from tropical cyclones may increase 20 % by 2100 (GFDL, 2013; Knutson et al., 2010).



Several studies about the effects of climate change on tropical cyclone intensity have been conducted for the Corpus Christi, TX area (Frey et al., 2010; Mousavi et al., 2011). Frey et al. (2010) conducted a study to determine how severe historical hurricanes would be if they were to occur in the current climate, and those predicted for the 2030s and 2080s. They found that, in all three climate scenarios, storm-surge flood depth, area of flood inundation, population affected, and

economic damages would all increase compared to the historical levels (Frey et al., 2010). In a follow-up study by Mousavi et al. (2011), the rise in storm-surge flood depth in response to global warming was found to be related to tropical cyclone intensification, measured in terms of central pressure. They found that sea level rise and tropical cyclone intensification contribute equally to increased flood depths (Mousavi et al., 2011). This second study indicates that flooding following severe hurricane events is likely to have a detrimental impact on highly populous coastal areas (Mousavi et al., 2011).

While there has been much study of the impact of tropical cyclones on coastal flooding, there has been little research on how these high-intensity precipitation events affect the hydrology of streams just inland of coastal areas. Further, few studies have focused on how inland flooding is likely to be altered with an extended hurricane season in the near future due to likely global climate change. This study investigates the potential increase in flooding risk with an extension of the hurricane season on four rivers in the Southeast United States. The goal of this study is to help determine how flooding

potential may change in the near future in order to elucidate the impact such changes may have on communities in the Southeast United States.

## 2 Study Areas

This study is focused in the Southeast United States, where tropical cyclone events occur more frequently, and where severe flooding following these events can have profound impacts on the prosperity of communities. Specifically, four river

basins (Neches, Pearl, Mobile, and Roanoke) were selected for analysis in this study (Fig. 1; Table 1). These four study basins were chosen to be in areas that experience tropical cyclones, and a high number of severe hurricanes. Gaging stations along these rivers were chosen to be inland of coastal areas so that tidal fluctuation and storm surge would not be factors when analyzing discharge, and far enough downstream to include as much of the study basins as possible. These four basins were selected to represent a range of sizes and geographic locations that exist throughout the Southeast United States.

United States Geological Survey (USGS) gages were used where data was available for the period extending from 1998-2014, the time frame analyzed in this study. In many cases USGS stream gages either did not have daily discharge data or did not have a long enough history of daily discharge data, or if sufficient daily discharge data was available, the location of




the gaging station was either too close to the coast where there were tidal fluctuations, or too far upstream in the catchment such that only a small fraction of the catchment was flowing to the gaging station. In these situations, Dartmouth Flood Observatory (DFO) satellite river gages where used (Brakenridge, De Groeve, Kettner, Cohen, and Nghiem, 2016).

## 3 Methods

### 3.1 Frequency and Timing of Tropical Cyclones

NOAA's HURDAT2 dataset (Landsea, Franklin, and Beven, 2015) was used to determine when tropical cyclones passed over the four study basins. For each tropical cyclone event on record, this dataset provides information on the year, month, day, time, latitude, longitude, maximum sustained wind speed (in knots), minimum pressure (in millibars), and several wind speed radii extents for points along a tropical cyclone's track (where points are spaced at 6-hour intervals). The data provided in the HURDAT2 dataset is downloadable in a text file format. A Python script was developed to extract the information provided in this database in order to create point shapefiles of tropical cyclone paths that could be analyzed in GIS. The paths of tropical cyclones between 1998 and 2014 were buffered to a width of 300 mi (~500 km), the average size of a tropical cyclone (Darby, Leyland, Kummu, Räsänen, and Lauri, 2013). Then, a selection by location procedure was used to determine which buffered tropical cyclones passed over each of the study basins. The latitudes and longitudes of the buffered points along tropical cyclone paths passing over the basins were then used to look up the corresponding dates each storm passed over each basin in the HURDAT2 dataset.

### 3.2 Determining Bankfull Discharge

Daily discharge data for the outlet of each of the study basins over the period from 1998-2014 was obtained from either the USGS or the DFO's Satellite River Discharge Measurements. The DFO sites provide daily measures of discharge since January 1, 1998 (Brakenridge, Cohen, Kettner, De Groeve, and Nghiem, 2012). Discharge is estimated from NASA and the Japanese Space Agency TRMM microwave data (Brakenridge et al., 2012). This dataset is particularly useful because it allows the user to place gaging stations at any location along world rivers. Using the daily discharge data obtained, the Log-Pearson Type III statistic (IACWD, 1982) was calculated for each basin. The Log-Pearson Type III statistic can be used to provide an "industry standard" of bankfull discharge for a river at a particular gaging station; times when discharge is greater than the bankfull discharge indicate the occurrence of a flood (IACWD, 1982). The bankfull discharge typically has a return period of 2.33 years (Waylen, 1991). In Kostaschuk et al.'s (2001) study of tropical cyclone floods in Fiji, the Log-Pearson

Type III statistic was found to more accurately represent their partial duration flood series than the Pareto distribution, even though it tended to slightly underestimate the largest flows.

The Log Pearson Type III statistic was calculated using maximum yearly discharge values from 1998-2014:

$$\log Q = \log \bar{Q} + K\sigma \tag{1}$$

where Q is the discharge of some return period, $\log \bar{Q}$ is the average of the log Q maximum discharge values, K is the frequency factor (found using the K frequency factor table, which is based upon return period and the skew coefficient), and $\sigma$ is the standard deviation of the log Q discharge values (OSU, 2005). The variance can be found using Eq. (2):

$$\sigma^2 = \frac{\sum_1^n (\log Q - \log \bar{Q})^2}{n-1} \tag{2}$$

where n is the number of maximum discharge values (i.e. the number of years) (OSU, 2005). The standard deviation can be found by taking the square root of the variance (OSU, 2005). The skew coefficient can be found using (OSU, 2005):

$$C_s = \frac{n \sum (\log Q - \log \bar{Q})^3}{(n-1)(n-2)(\sigma^3)} \tag{3}$$

The bankfull discharge was calculated using a return period of 2.33, following Waylen (1991). This study focuses on discharge and flooding susceptibility at the mouths/outlets of the four study basins inland of coastal areas. It is assumed that conditions at these areas are indicative of much of the main river stem.

### 3.3 Analyzing the Effects of an Extended Hurricane Season on Flooding Susceptibility

An analysis was performed to determine how many days from 1998-2014 during the hurricane season would have been at risk of flooding were an average tropical cyclone to have occurred on any given day. For each tropical cyclone in each basin from 1998-2014 the discharge the day before the event was compared to the peak discharge in order to determine the increases in discharge due to the tropical cyclones. Within each basin, these increases in discharge were averaged to determine the average increase in discharge due to a tropical cyclone.



For each individual day from 1998-2014 between the months of June-November, the daily discharge in the Neches River was increased by the average increase in discharge due to a tropical cyclone experienced by the Neches Basin. This increased discharge due to a hypothetical average tropical cyclone was compared with the bankfull discharge value on each individual day for the Neches River. A day with a hypothetical discharge greater than bankfull discharge indicates that the

Neches River likely would have flooded on this day if an average tropical cyclone were to have impacted this basin. Similar analyses were conducted for the Pearl, Mobile, and Roanoke basins. This procedure allows for an analysis of how many days within each basin would have been likely to flood were an average tropical cyclone to have occurred on any given day from 1998-2014 between the months of June-November.

The above methodology was then repeated with an extended hurricane season of May-December. This procedure allows

for an analysis of how many days within each basin would have been likely to flood were an average tropical cyclone to have occurred on any given day from 1998-2014 between the months of May-December. The current hurricane season is June-November (Dwyer et al., 2015), thus May-December was chosen to see what effect an extension of the hurricane season by one month on either side of the season might have on flooding in these basins. A 1-month extension was considered because several May (1 month outside the current hurricane season) tropical cyclones have impacted the Roanoke

Basin in recent years. NOAA's HURDAT2 dataset also indicates the occurrence of some May, as well as some December, Atlantic tropical cyclones. This data was then compared to the percentage of days susceptible to tropical cyclone-induced flooding in current hurricane season.

## 4 Results

### 4.1 Tropical Cyclone Frequency and Timing

From 1998-2014 (17 years), 15 tropical cyclones impacted the Neches Basin, 28 impacted the Pearl Basin, 30 impacted the Mobile Basin, and 36 impacted the Roanoke Basin. The number of tropical cyclones impacting each basin each year has not been constant over the period of study. The years 2004 and 2005 had high numbers of storms in every basin, and in recent years there have been very few storms. For example, in 2004 and 2005 most basins experienced 2 or more tropical cyclones, while in 2013 and 2014 only the Roanoke basin was impacted by tropical cyclones (and only 1 in each year). Most

notably, almost all tropical cyclones impacting these four basins occur during low-discharge seasons, when flood risk is minimal (Figs. 2 and 3).





### 4.2 Effects of an Extended Hurricane Season on Flooding Susceptibility

The current hurricane season is defined as starting in June and ending in November (Dwyer et al., 2015). More recent years (2007, 2009, 2012) have shown some tropical cyclones first occurring in May, before the "official" hurricane season, in the Roanoke Basin (Figs. 2d and 3d). NOAA's HURDAT2 dataset also indicates the occurrence of some May, as well as

some December, Atlantic tropical cyclones. On average, tropical cyclones increased discharge (calculated from the difference between peak discharge and discharge the day before the storm) by 97.85 cms on the Neches River, 226.71 cms on the Pearl River, 787.25 cms on the Mobile River, and 101.26 cms on the Roanoke River (Table 2). The average percent increase in discharge following a tropical cyclone impact in all four rivers was 92 % (Table 2).

From 1998-2014, during the current hurricane season months of June-November, on about 30 days (out of 3,111 days in

the 1998-2014 June-November hurricane seasons) the Neches River would be above bankfull discharge and at risk to flooding were an average tropical cyclone to occur (Fig. 4a; Table 2). That is, about 0.96 % of days are susceptible to potential flooding were an average tropical cyclone to occur (Table 2). Similarly, the flood risk was found to be about 39 days (or 1.25 % of the time) for the Pearl River, about 10 days (or 0.32 % of the time) for the Mobile River, and about 50 days (or 1.61 % of the time) for the Roanoke River (Figs. 5a, 6a, and 7a) (Table 2). The average susceptibility for potential

tropical cyclone-induced flooding for all four rivers was about 32 days 1 (or 1.04 % of the time) (Table 2).

The extended hurricane season of May-December scenario showed greater flooding risk for all four of the rivers. The flood risk was found to be about 44 days (or 1.06 % of the time) for the Neches River, about 50 days (or 1.20 % of the time) for the Pearl River, about 28 days (or 0.67 % of the time) for the Mobile River, and about 84 days (or 2.02 % of the time) for the Roanoke River (Figs. 4b, 5b, 6b, and 7b) (Table 2). The average for all four rivers was 1.24 % of the time (or about 52

days) (Table 2). On average, the extended hurricane scenario led to about 20 more days per basin that likely would be at risk to a flood were the average tropical cyclone to occur (Table 2). Over the 17 seasons we investigated, this is about a 63 % increase in the number of days at risk to flooding if an average tropical cyclone were to impact these basins. From a yearly perspective, this is an increase from 1.9 days/yr to 3.1 days/yr that are at risk of tropical cyclone-induced flooding.

### 5 Discussion

The results of this study reveal that most tropical cyclones impacting these four basins occur during September, or the middle of the low discharge season (Figs. 2 and 3). The current hurricane season is defined as beginning in June and lasting through November (Dwyer et al., 2015), which coincides primarily with the low discharge seasons of the four basins



analyzed in this study. Thus, tropical cyclones rarely cause flood events on these rivers, even though they bring high amounts of precipitation, because they occur primarily during the low discharge season. This is in contrast to tropical cyclones in Southeast Asia, for example, which are frequent during the monsoon season, causing widespread inland flooding (Darby et al., 2013). Some May tropical cyclones have already occurred in the Roanoke Basin during 2007, 2009, and 2012, and NOAA's HURDAT2 dataset contains other May and December tropical cyclones occurring in the Atlantic Ocean. This suggests that while tropical cyclones rarely led to inland flooding from 1998-2014 in the four Southeast basins analyzed in this study, a future extension of the hurricane season, such that it encroaches upon the high discharge season in these rivers, has the potential to considerably enhance flooding risks.

It was the goal of this study to determine the increase in flood risk with a May-December hurricane season versus the June-November hurricane season. As described in the results above, a May-December hurricane season was found to increase flood risk to the average tropical cyclone in all of the study basins. Adding the months of May and December increases the number of days per year in the hurricane season from 183 to 245; this is about a 34 % increase in the number of days during the year that fall within the hurricane season. For the Neches River, 14 additional days (a 47 % increase) would have been at risk of flooding were the average tropical cyclone to occur on any given day from 1998-2014 with an extended May-December hurricane season (Table 2). Similarly, 11 additional days (a 28 % increase) would have been at risk of flooding on the Pearl River, 18 additional days (a 180 % increase) on the Mobile River, and 34 additional days (a 68 % increase) on the Roanoke River (Table 2). Thus, for just a 34 % increase in the length of the hurricane season there was, on average, an 81 % increase in the number of days at risk to a tropical-cyclone-induced flood along these Southeast rivers (Table 2). When averaged over the 17-year period analyzed in this study, the number of days at risk of tropical cyclone-induced flooding increases from 1.9 days/yr to 3.1 days/yr. While 3 day/yr may not seem like a high risk factor, it not only represents a 63 % increase, but it is also a very conservative number, not taking into account predicted enhancements in the intensity and/or frequency of future tropical cyclones (Bronstert et al., 2002; Frey et al., 2010; Greenough et al., 2001; Irish and Resio, 2013; Irish et al., 2014; Kostaschuk et al., 2001; Mousavi et al., 2011; Ouellet et al., 2012). Increases in tropical cyclone intensity and frequency in the near future is likely to greatly increase the number of days at risk to flooding in the four basins analyzed in this study, which will be the focus of a future study.

The timing of the hurricane season in relation to the high and low discharge seasons is crucial to understanding flooding risk following tropical cyclones on these rivers. The Mobile and Roanoke rivers showed the greatest increase in flooding risk following the average tropical cyclone (68 % and 180 % respectively) in an extended May-December hurricane season; discharges increase significantly more between November and December for the Mobile and Roanoke rivers (Figs. 2c and



2d) than for the Neches and Pearl rivers (Figs. 2a and 2b). The Pearl River showed the least increase in flooding risk following the average tropical cyclone (28 %) in an extended May-December hurricane season; while the Neches, Mobile, and Roanoke rivers tend to have slightly higher discharges in May than in June, discharges in the Pearl River are slightly lower in May than June. Thus, this study reveals that flooding risk following tropical cyclones not only is expected to

increase were the hurricane season to be extended due to global climate warming, but also that this increase will not be uniform across the Southeast United States. Rivers whose high discharge seasons include the months of May and December, such as the Mobile and Roanoke rivers, are likely to be most affected by a lengthened hurricane season.

The main limitation of this study is its use of average statistics. Future work could extend this study to look at increase in flood risk not only due to the average tropical cyclone, but the full range of tropical cyclones a basin is likely to experience

(the tropical cyclone with the maximum increase in discharge, the tropical cyclone with the minimum increase in discharge, etc.). For instance, given that tropical cyclones are likely to intensify (Bronstert et al., 2002; Frey et al., 2010; Greenough et al., 2001; Irish and Resio, 2013; Irish et al., 2014; Kostaschuk et al., 2001; Mousavi et al., 2011; Ouellet et al., 2012), flooding risk in an extended hurricane season likely could be greater than the results presented in this paper. Further, more explicit modeling of future tropical cyclone dynamics using a stochastic approach, rather than average statistics, could

potentially produce a more robust understanding of the effects of future climate dynamics on flood susceptibility. Because the high discharge season varies from basin to basin, extending this study to other basins along the East and Gulf coasts would allow for a fuller understanding of which areas in the Southeast United States are likely to be much more at risk to flooding following tropical cyclones due to an extension of the hurricane season in response to global climate warming.

**6 Conclusions**

Most tropical cyclones impacting the Neches, Pearl, Mobile, and Roanoke river basins from 1998-2014 occurred during the low discharge seasons on these rivers, and caused little flooding. Although the current hurricane season is June-November, some May and December tropical cyclones have occurred in recent years. An extension of the hurricane season to May-December would likely increase flooding risk on the four rivers analyzed in this study by 28-180 % depending on the amount of overlap between the high discharge season with the hurricane season. On the Neches River, 14 more days (a 47

% increase) from 1998-2014 in the months May-December would have been at risk of flooding if an average tropical cyclone had occurred on any given day than during the current hurricane season months of June-November. The Pearl, Mobile, and Roanoke basins had similar increases of 11 more days (a 28 % increase), 18 more days (a 180 % increase), and



34 more days (a 68 % increase) respectively. On average, about 20 more days per basin would be a risk to a tropical-cyclone-induced flood in an extended May-December hurricane season scenario for the 17 hurricane seasons we investigated. On a yearly scale, an extended hurricane season may increase the number of days at risk of flooding in response to the average tropical cyclone from 1.9 days/yr to 3.1 days/yr, a 63 % increase. More work is needed to extend

this analysis to all river basins in the United States that are affected by tropical cyclones in order to provide a better sense of how inland flooding following tropical cyclones in the United States is likely to change were the hurricane season to lengthen in the near future due to global climate warming.

## Acknowledgements

We wish to thank Dr. Jason Senkbeil and Dr. Peter Waylen for their guidance in this research. We are thankful for the

help of Dr. G. Robert Brakenridge for his help in installing river gages for some of the study basins with the Dartmouth Flood Observatory. And lastly, thank you to The University of Alabama for funding portions of this research.

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



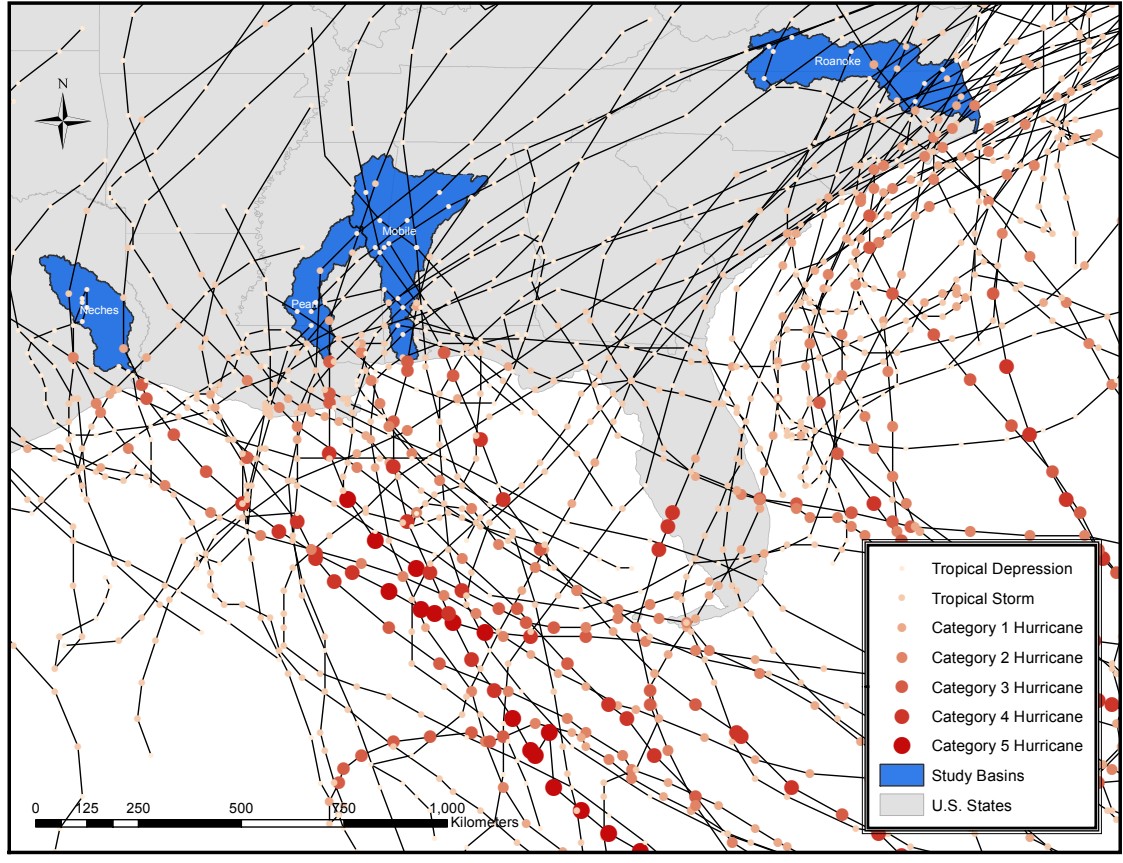

**Figure 1: Location of the study basins analyzed in this study (blue); colored dots represent points along the tracks of all tropical cyclones since 1998 that impacted the study basins, where the color/size of the dot indicates the severity of the storm at that location (see legend). (HURDAT2, NHD, ESRI)**



**Table 1: Location and size of the four study basins.**

| River Basin | Near | Latitude | Longitude | Basin Size |
|---|---|---|---|---|
| **Neches** | Silsbee/Evadale, TX | 30.374 | -94.094 | 25,117 km$^2$ |
| **Pearl** | Slidell, LA | 30.374 | -89.774 | 22,894 km$^2$ |
| **Mobile** | Mt. Vernon, AL | 31.094 | -87.974 | 110,955 km$^2$ |
| **Roanoke** | Williamston, NC | 35.864 | -76.904 | 25,963 km$^2$ |





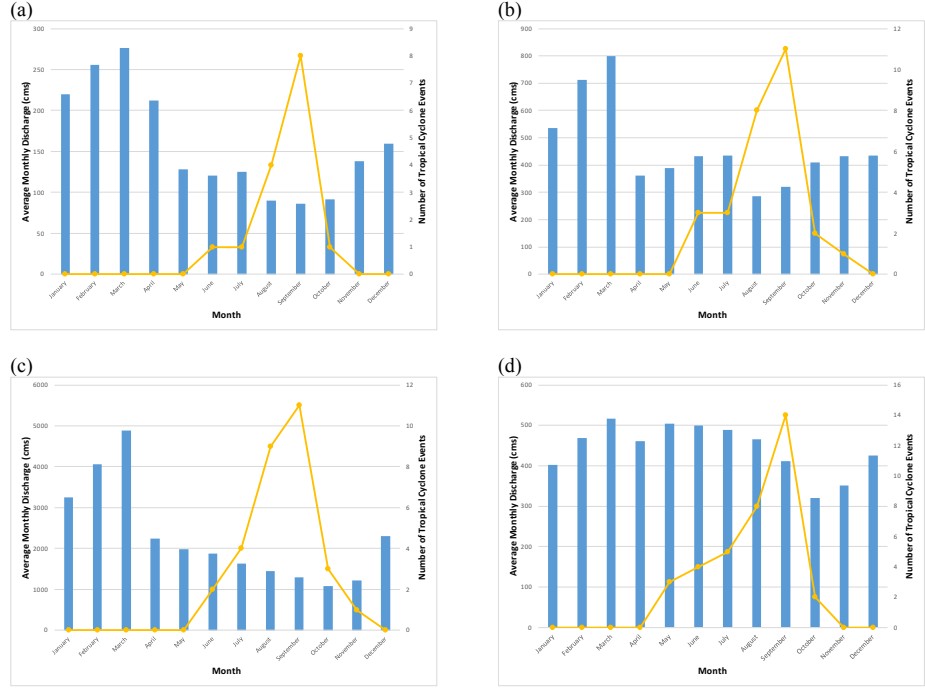

**Figure 2: Comparison of average monthly discharge (blue bars) with the number of tropical cyclones occurring each month (yellow line) from 1998-2014 for the Neches (a), Pearl (b), Mobile (c), and Roanoke (d) basins.**





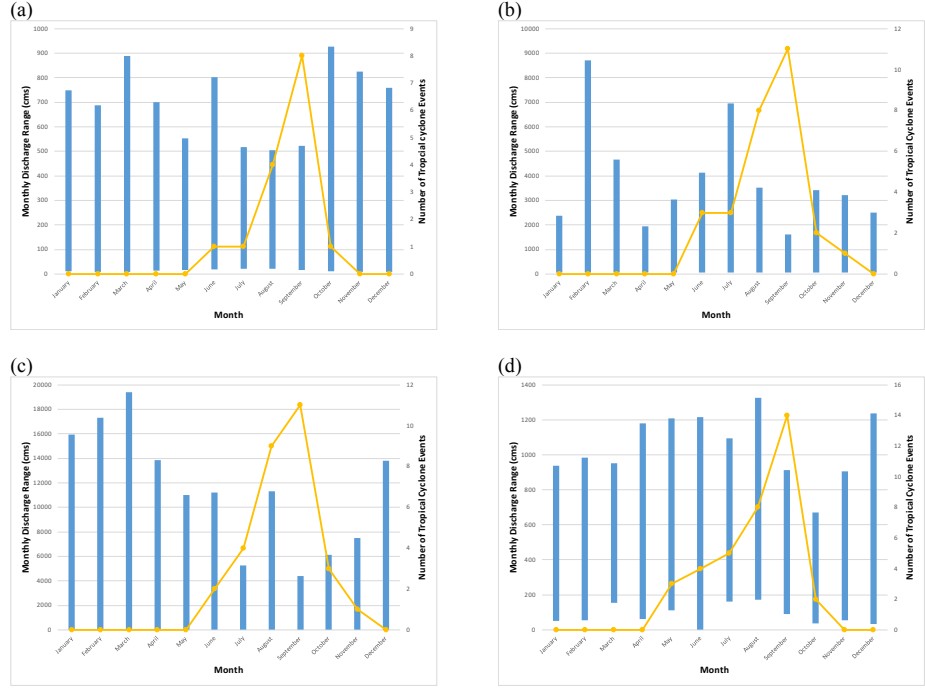

**Figure 3: Comparison of monthly discharge maximum/minimum range (blue bars) with the number of tropical cyclones occurring each month (yellow line) from 1998-2014 in the Neches (a), Pearl (b), Mobile (c), and Roanoke (d) basins.**





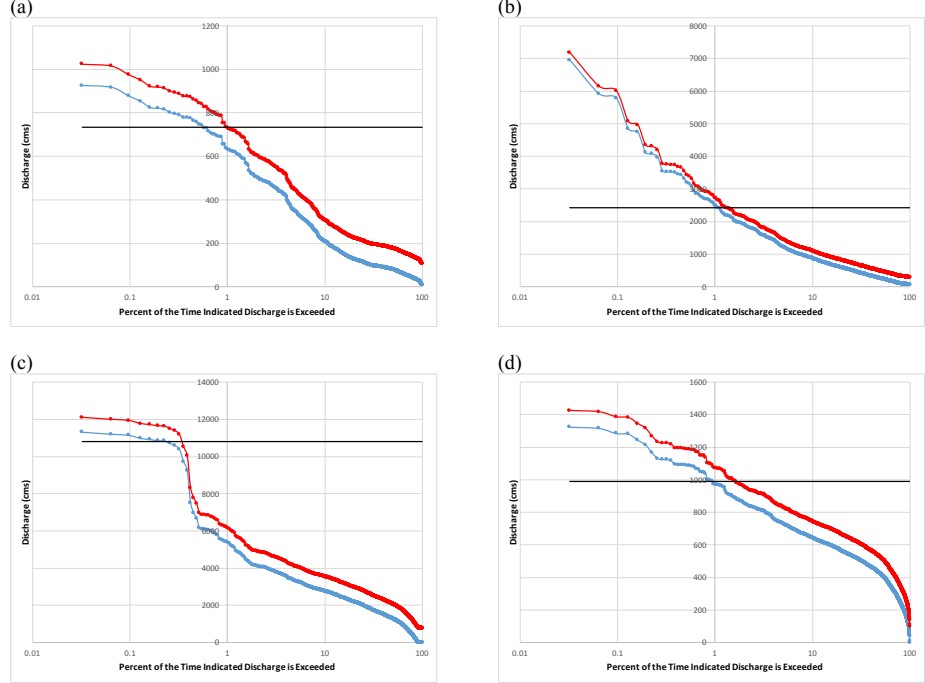

**Figure 4: Bankfull discharge (black lines), flow duration curves (blue curves), and flow duration curves with discharge increased due to the average tropical cyclone (red curves) for the current hurricane season on the Neches (a), Pearl (b), Mobile (c), and Roanoke (d) rivers.**





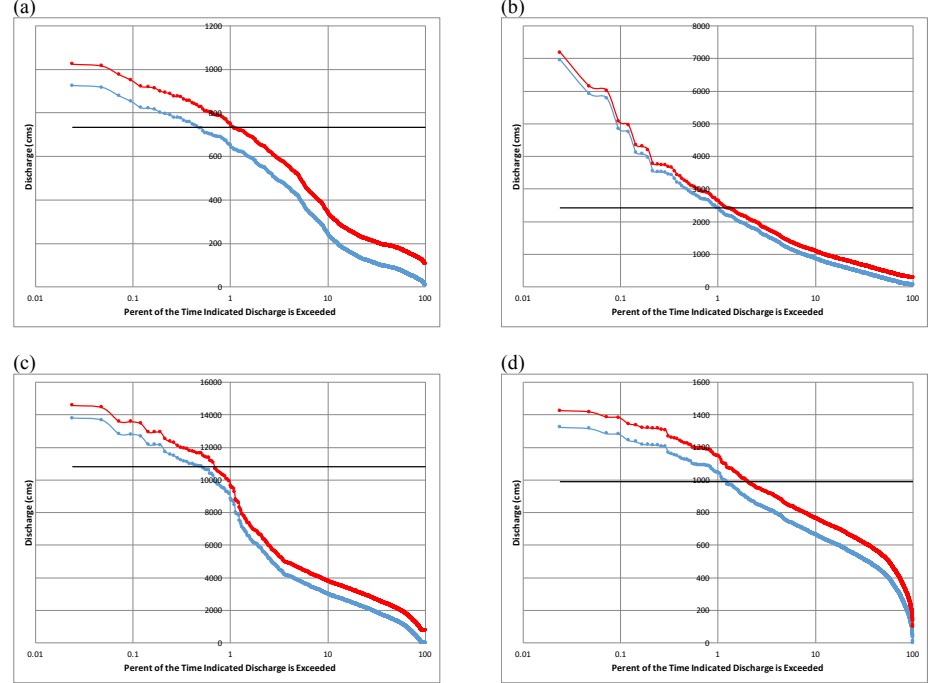

**Figure 5: Bankfull discharge (black lines), flow duration curves (blue curves), and flow duration curves with discharge increased due to the average tropical cyclone (red curves) for an extended May-December hurricane season on the Neches (a), Pearl (b), Mobile (c), and Roanoke (d) rivers.**



**Table 2: Flooding risk from 1998-2014 for the four study basins with the current hurricane season and with an extended May-December hurricane season.**

| Basin | Increase in Discharge due to Average Tropical Cyclone | Days at risk with June-Nov. Season (% of time period) | Days at risk with May-Dec. Season (% of time period) | Increase in Risk with Extended Season |
|---|---|---|---|---|
| Neches | 97.85 cms | 30 (0.96 %) | 44 (1.06 %) | + 14 days (47 % increase) |
| Pearl | 226.71 cms | 39 (1.25 %) | 50 (1.20 %) | + 11 days (28 % increase) |
| Mobile | 787.25 cms | 10 (0.32 %) | 28 (0.67 %) | + 18 days (180 % increase) |
| Roanoke | 101.26 cms | 50 (1.61 %) | 84 (2.02 %) | + 34 days (68 % increase) |
| **Average** | **92 % increase** | **32 (1.04 %)** | **52 (1.24%)** | **63 % increase in # of days at risk** |