# Peer review of "The influence of an extended Atlantic hurricane season on inland flooding potential in the Southeast United States"

_Natural Hazards and Earth System Sciences, 2016_

## Referee Comment (RC1) · Anonymous Referee #1 · 9 Nov 2016

The premise of this paper is simple and the methods are straightforward, and the topic is interesting. The authors quantify the change in flood risk in four southeastern U.S. drainage basins under the assumption that the Atlantic hurricane season would increase by one month at the beginning and one month at the end of the currently delineated tropical cyclone season. Such research is placed in the proper theoretical context, as the expected continued warming would leave ocean temperatures warm enough to sustain a tropical system for a larger number of months per year. Furthermore, the authors do a proper job, without getting too bogged down in tangential points, of introducing the reader to the somewhat conflicted literature on whether it would be the frequency and/or intensity of tropical cyclones that would increase under

such warming. A minor point: I think that the wording is a bit strong on Page 3, Line 13, where the authors say, "This hypothesis was refuted by..." – at a minimum, a hypothesis can't be refuted when one of the papers doing the refuting was written before the opposing papers, but more importantly, I think the jury is still out one which hypothesis is correct. That point notwithstanding, I like the theoretical set-up for the paper.

The primary theoretical/methodological weakness of the paper is the failure to account for synergistic effects of interactions between May or December tropical cyclones with extratropical systems. We all saw in 2012 (i.e., Sandy) how such interactions can cause greatly increased precipitation totals. At a bare minimum, the authors need to acknowledge this as a major weakness of the study.

The chief non-theoretical/methodological weakness is that the paper could have delivered the same message in perhaps 60% of the words. Even though I generally enjoyed reading the manuscript, I continually found myself a bit frustrated and thinking, "not again?!?" when I read repetitious text or text that was unnecessary. If the text were tightened fairly significantly, I'm sure that I and many others would find the paper to be a nice contribution to the literature. I attach a marked-up version of the manuscript in the hope that this will assist the authors as they tighten the manuscript.

One other comment: Please insert the word "Atlantic" in the title and elsewhere in the text, to show that your study only considers one of the world's tropical cyclone-vulnerable areas.

Please also note the supplement to this comment:
http://www.nat-hazards-earth-syst-sci-discuss.net/nhess-2016-320/nhess-2016-320-RC1-supplement.pdf
* * *

---

## Referee Comment (RC2) · R. Rohli (Referee) · 17 Nov 2016

I read the corresponding author's comments in response to my comments, and even though I didn't re-read the paper, I feel that the authors have addressed my concerns.

Good job on this work.
* * *

---

## Author Comment (AC1) · 17 Nov 2016

We thank the reviewer for the insightful comments. We have addressed all of them in the revised manuscript. Below we provide a point-by-point response.

"The premise of the paper is simple and the methods are straightforward, and the topic is interesting. The authors quantify the change in flood risk in four southeastern U.S. drainage basins under the assumption that the Atlantic hurricane season would increase by one month at the beginning and one month at the end of the currently delineated tropical cyclone season. Such research is placed in the proper theoretical context, as the expected continued warming would leave ocean temperatures warm enough to sustain a tropical system for a larger number of months per year. Furthermore, the authors do a proper job, without getting too bogged down in tangential points, of introducing the reader to the somewhat conflicted literature on whether it would be the frequency and/or intensity of tropical cyclones that would increase under such warming."

Thank you.

"A minor point: I think that the wording is a bit strong on Page 3, Line 13, where the authors say, 'This hypothesis was refuted by. . .' – at a minimum, a hypothesis can't be refuted when one of the papers doing the refuting was written before the opposing papers, but more importantly, I think the jury is still out on which hypothesis is correct. That point notwithstanding, I like the theoretical set-up for the paper."

The sentences in question were amended as follows:

Major hurricanes, those that are Category 3 or higher on the Saffir-Simpson scale, are the most likely to intensify (Frey et al., 2010; Mousavi et al., 2011), but there is some debate about changes in tropical cyclone frequency. Some research predicts that tropical cyclone frequency will increase (e.g. Greenough et al., 2001; Ouellet et al., 2012), while other research suggests that tropical cyclones are likely to intensify with global climate warming, but occur less frequently (e.g. Irish and Resio, 2013; Kostaschuk et al., 2001).

"The primary theoretical/methodological weakness of the paper is the failure to account for synergistic effects of interactions between May or December tropical cyclones with extratropical systems. We all saw in 2012 (i.e., Sandy) how such interactions can cause greatly increased precipitation totals. At a bare minimum, the authors need to acknowledge this as a major weakness of the study."

We added this as a weakness of our paper (page 8, starting in line 18):

Further, this research does not consider synergistic effects due to the potential inter-play between May and/or December tropical cyclones and mid-latitude cyclones, which

could increase precipitation and flooding risk even further.

"The chief non-theoretical/methodological weakness is that the paper could have delivered the same message in perhaps 60% of the words. Even though I generally enjoyed reading the manuscript, I continually found myself a bit frustrated and thinking, 'not again?!?' when I read repetitious text or text that was unnecessary. If the text were tightened fairly significantly, I'm sure that I am many others would find the paper to be a nice contribution to the literature. I attach a marked-up version of the manuscript in the hope that this will assist the authors as they tighten the manuscript."

Thank you. We incorporated all of the suggestions in the marked-up version in the revised manuscript.

"One other comment: Please insert the word 'Atlantic' in the title and elsewhere in the text, to show that your study only considers one of the world's tropical cyclone-vulnerable areas."

Done!

Please also note the supplement to this comment:
http://www.nat-hazards-earth-syst-sci-discuss.net/nhess-2016-320/nhess-2016-320-AC1-supplement.pdf

**Supplement:**

[revised manuscript text omitted]

---

## Referee Comment (RC3) · Anonymous Referee #2 · 29 Dec 2016

This premise of this study is simple, yet in its simplicity it answers a fundamental and important question. The methodology is well written and easy to understand which allows the reader to fully engage and understand the reasoning on how and why this important science question posed can be answered.

Yet, there are a few minor tweaks that could really help this paper. Like the first reviewer, I think the writing could be more succinct, especially in the introduction section. For example, paragraphs 5 and 6 in this section could be merged to avoid some repeating. More details can be found in the marked up pdf.

I would like to see further justification for the selection of the study sites and/or examples of flood events and their characteristics. For example, when was the last large

flood in each basin and was it associated with a tropical cyclone? This will particular help the reader not familiar with the basins in question. This could be in a simple table to aid readability.

In the discussion section you have outlined some limitations, but I think you could add that the intensity of the storms could be lower in May and December than the average statistic used. This could mean that the expected increase could be less as well as greater to what you have presented in the paper. I believe in the further study these issues will be addressed but I think this limitation should at least be mentioned here.

Lastly, a brief discussion about the uncertainty associated with the DFO satellite river discharge measurement sites needs to be included.

Please also note the supplement to this comment:
http://www.nat-hazards-earth-syst-sci-discuss.net/nhess-2016-320/nhess-2016-320-RC3-supplement.pdf

[Figure]

**Supplement:**

[revised manuscript text omitted]

---

## Author Comment (AC2) · 12 Jan 2017

We thank the reviewer for the insightful comments. We have addressed all of them in the revised manuscript. Below we provide a point-by-point response.

"The premise of this study is simple, yet in its simplicity it answers a fundamental and important question. The methodology is well written and easy to understand, which allows the reader to fully engage and understand the reasoning on how and why this important science question posed can be answered."

Thank you.

"Yet, there are a few minor tweaks that could really help this paper. Like the first re-

viewer, I think the writing could be more succinct, especially in the introduction section. For example, paragraphs 5 and 6 in this section could be merged to avoid some repeating. More details can be found in the marked up pdf."

The changes suggested in the marked up pdf were made. Please see the revised manuscript.

"I would like to see further justification for the selection of the study sites and/or examples of flood events and their characteristics. For example, when was the last large flood in each basin and was it associated with a tropical cyclone? This will particularly help the reader not familiar with the basins in question. This could be in a simple table to aid readability."

The following explanation was added in the first paragraph of the "Study Areas" section:

This research is focused in the south-eastern United States, where tropical cyclone events occur quite frequently, and where severe flooding following these events can have profound impacts on the prosperity of communities. Specifically, four river basins (Neches, Pearl, Mobile, and Roanoke) were selected for analysis (Fig. 1; Table 1). These four basins were chosen to be in areas that experience tropical cyclones, and a high number of severe hurricanes. Currently, tropical cyclones impacting these four basins rarely cause flooding. As is shown later in this paper, this is primarily due to the overlap of the current hurricane season with the low discharge seasons on these four rivers. However, an extension of the hurricane season, such that it encroaches upon the high discharge seasons on these rivers, could likely lead to increases in flooding following tropical cyclones that impact these basins.

"In the discussion section you have outlined some limitations, but I think you could add that the intensity of the storms could be lower in May and December than the average statistic used. This could mean that the expected increase could be less as well as greater to what you have presented in the paper. I believe in the further study these issues will be addressed, but I think this limitation should at least be mentioned here."

We added this as a limitation to our study on the line suggested in marked up pdf.

"Lastly, a brief discussion about the uncertainty associated with the DFO satellite river discharge measurement sites needs to be included."

The following sentences about the DFO satellite river gages were added:

Brakenridge et al. (2012) tested the accuracy of DFO satellite river discharge measurements and reported regression r2 values > 0.6. They also provide a site specific "Quality Assessment" which, for sites in the United States, is based on calculating the Nash-Sutcliffe (NS) statistics for the DFO site and near gaging station hydrographs (Brakenridge et al., 2015). For the Mobile River site, for example, the DFO "Quality Assessment" ranking is 2 (Fair), which means that the NS statistics were > 0.44. However, since both bankfull and time series discharge are esti-mated from the same source in this study, while the absolute value may somewhat differ from the actual discharge, temporal trends and fluctuation magnitude were found to be well captured. This is clearly evident in the Mobile River DFO site (http://floodobservatory.colorado.edu/SiteDisplays/467.htm).

Please also note the supplement to this comment:
http://www.nat-hazards-earth-syst-sci-discuss.net/nhess-2016-320/nhess-2016-320-AC2-supplement.pdf

**Supplement:**

[revised manuscript text omitted]